# ADDITIVE POISSON PROCESS:
# LEARNING INTENSITY OF HIGHER-ORDER
# INTERACTION IN POISSON PROCESSES

## ABSTRACT

We present the *Additive Poisson Process* (APP), a novel framework that can model the higher-order interaction effects of the intensity functions in Poisson processes using projections into lower-dimensional space. Our model combines the techniques from *information geometry* to model higher-order interactions on a statistical manifold and in *generalized additive models* to use lower-dimensional projections to overcome the effects from the curse of dimensionality. Our approach solves a convex optimization problem by minimizing the KL divergence from a sample distribution in lower-dimensional projections to the distribution modeled by an intensity function in the Poisson process. Our empirical results show that our model is able to use samples observed in the lower dimensional space to estimate the higher-order intensity function with extremely sparse observations.

## 1 INTRODUCTION

The Poisson process is a counting process used in a wide range of disciplines such as spatial-temporal sequential data in transportation (Zhou et al., 2018; 2021), finance (Ilalan, 2016) and ecology (Thompson, 1955) to model the arrival rate by learning an intensity function. For a given time interval, the integral of the intensity function represents the average number of events occurring in that interval.

The intensity function can be generalized to multiple dimensions. However, for most practical applications, learning the multi-dimensional intensity function becomes a challenge because of the sparse observations. Despite the recent advances of Poisson processes, most current Poisson process models are unable to learn the intensity function of a multi-dimensional Poisson process. Our research question is, "Are there any good ways of approximating the high dimensional intensity function?" Our proposed model, the *Additive Poisson Process* (APP), provides a novel solution to this problem.

Throughout this paper, we will use a running example in a spatial-temporal setting. Say we want to learn the intensity function for a taxi to pick up customers at a given time and location. For this setting, each event is multi-dimensional; that is, $(x_i, y_i, W_i)$, where a pair of $x_i$ and $y_i$ represents two spatial coordinates and $W_i$ represents the day of the week. In addition, observation time is associated with each component of the event, which is represented as $\mathbf{t}_i = (t_{x_i}, t_{y_i}, t_{W_i})$. In this example, it is reasonable to assume $t_{x_i} = t_{y_i} = t_{W_i}$ as they are observed simultaneously. For any given location or time, we can expect at most a few pick-up events, which makes it difficult for any model to learn the low-valued intensity function. Figure 2b visualizes this problem. In this problem setup, if we would like to learn the intensity function at a given location $(x, y)$ and day of the week $W$, the naïve approach would be to learn the intensity at $(x, y, W)$ directly from observations. This is extremely difficult because there could be only few events for a given location and day.

However, there is useful information in lower-dimensional space; for example, the marginalized observations at the location $(x_i, y_i)$ across all days of the week, or on the day $W_i$ at all locations. This information can be included into the model to improve the estimation of the joint intensity function. Using the information in lower-dimensional space provides a structured approach to include prior information based on the location or day of the week to improve the estimation of the joint intensity function. For example, a given location could be a shopping center or a hotel, where it is common for taxis to pick up passengers, and therefore we expect more passengers at this location. There

could also be additional patterns that could be uncovered based on the day of the week. We can then use the observations of events to update our knowledge of the intensity function.

Previous approaches, such as kernel density estimation (KDE) (Rosenblatt, 1956), are able to learn the joint intensity function by using information in lower dimensions. However, KDE suffers from the curse of dimensionality, which means that it requires a large size of samples to build an accurate model. In addition, the complexity of the model expands exponentially with respect to the number of dimensions, which makes it infeasible to compute. Bayesian approaches, such as using a mixture of beta distributions with a Dirichlet prior (Kottas, 2006; Kottas & Sansó, 2007) and Reproducing Kernel Hilbert Space (RKHS) (Flaxman et al., 2017), have been proposed to quantify the uncertainty for the intensity function. However, these approaches are often non-convex, making it difficult to obtain the globally optimal solution. Besides, if observations are sparse, it is hard for these approaches to learn a reasonable intensity function. Additional related work about Bayesian inference for Poisson processes and Poisson factorization can be found in Appendix A.

In this paper, we propose a novel framework to learn the higher-order interaction effects of intensity functions in Poisson processes. Our model combines the techniques introduced by Luo & Sugiyama (2019) to model higher-order interactions between Poisson processes and by Friedman & Stuetzle (1981) in *generalized additive model*s to learn the joint intensity function using samples in a lower dimensional space. Our proposed approach is to decompose a multi-dimensional Poisson process into lower-dimensional representations. For example, we have points $(x_i)_{i=1}^N$ in the $x$-dimension and $(y_i)_{i=1}^N$ in the $y$-dimension. Such data in the lower-dimensional space can be used to improve the estimation of the joint intensity function. This is different from the traditional approaches where only the joint occurrence of events is used to learn the joint intensity.

We first show the connection between generalized additive models and Poisson processes, and then provide the connection between generalized additive models and the *log-linear model* (Agresti, 2012), which has a well-established theoretical background in information geometry (Amari, 2016). We draw parallels between the formulation of the generalized additive models and the binary log-linear model on a partially ordered set (poset) (Sugiyama et al., 2017). The learning process in our model is formulated as a convex optimization problem to arrive at a unique optimal solution using natural gradient, which minimizes the Kullback-Leibler (KL) divergence from the sample distribution in a lower-dimensional space to the distribution modeled by the learned intensity function. This connection provides remarkable properties to our model: the ability to learn higher-order intensity functions using lower-dimensional projections, thanks to the *Kolmogorov-Arnold representation theorem*. This property makes it advantageous to use our proposed approach for cases where there are no observations, missing samples, or low event rates. Our model is flexible because it can capture the interaction effects between events in a Poisson process as a partial order structure in the log-linear model and the parameters of the model are fully customizable to meet the requirements of the application. Our empirical results show that our model effectively uses samples projected onto a lower dimensional space to estimate the higher-order intensity function. More importantly, our model is also robust to various sample sizes.

## 2    FORMULATION

We start this section by introducing the technical background in the Poisson process and its extension to a multi-dimensional Poisson process. We then introduce the Generalized Additive Model (GAM) and its connection to the Poisson process. This is followed by presenting our novel framework, called Additive Poisson Process (APP), which is our main technical contribution and has a tight link to the Poisson process modeled by GAMs. We show that the learning of APP can be achieved via convex optimization using natural gradient.

The Poisson process is characterized by an intensity function $\lambda : \mathbb{R} \to \mathbb{R}$. An inhomogeneous Poisson process is an extension of a homogeneous Poisson process, where the arrival rate changes with time. The process with time-changing intensity $\lambda(t)$ is defined as a counting process $\mathbb{N}(t)$, which has an independent increment property. For any time $t \geq 0$ and infinitesimal interval $\delta \geq 0$, the probability of events count is $p(\mathbb{N}(t+\delta)-\mathbb{N}(t) = 0) = 1-\delta\lambda(t)+o(\delta)$, $p(\mathbb{N}(t+\delta)-\mathbb{N}(t) = 1) = \delta\lambda(t)+o(\delta)$, and $p(\mathbb{N}(t+\delta)-\mathbb{N}(t) \geq 2) = o(\delta)$, where $o(\cdot)$ denotes little-o notation (Daley & Vere-Jones, 2007). This formulation can be generalized into the (inhomogeneous) multi-dimensional Poisson process, which can be used to learn the intensity function $\lambda : \mathbb{R}^D \to \mathbb{R}$ from a realization

of timestamps $\mathbf{t}_1, \mathbf{t}_2, \ldots, \mathbf{t}_N$ with $\mathbf{t}_i \in [0, T]^D$. Each $\mathbf{t}_i$ is the time of occurrence for the $i$-th event across $D$ dimensions and $T$ is the observation duration. The likelihood for the multi-dimensional Poisson process (Daley & Vere-Jones, 2007) is given by

$$p\left(\{\mathbf{t}_i\}_{i=1}^N \mid \lambda(\mathbf{t})\right) = \exp\left(-\int \lambda(\mathbf{t})dt\right) \prod_{i=1}^N \lambda(\mathbf{t}_i), \tag{1}$$

where $\mathbf{t} = (t^{(1)}, \ldots, t^{(D)}) \in \mathbb{R}^D$. We define the functional prior on $\lambda(\mathbf{t})$ as

$$\lambda(\mathbf{t}) := g\left(f(\mathbf{t})\right) = \exp\left(f(\mathbf{t})\right). \tag{2}$$

The function $g(\cdot)$ is a positive function to guarantee the non-negativity of the intensity which we choose to be the exponential function, and our objective is to learn the function $f(\cdot)$. The log-likelihood of the multi-dimensional Poisson process with the functional prior is described as

$$\log p\left(\{\mathbf{t}_i\}_{i=1}^N \mid \lambda(\mathbf{t})\right) = \sum_{i=1}^N f(\mathbf{t}_i) - \int \exp\left(f(\mathbf{t})\right) dt. \tag{3}$$

In the following sections, we introduce the *generalized additive models* and propose to model it by the *log-linear model* to learn $f(\mathbf{t})$ and the normalizing term ($\int \exp\left(f(\mathbf{t})\right) dt$).

## 2.1 GENERALIZED ADDITIVE MODEL

In this section, we present the connection between Poisson processes and the Generalized Additive Model (GAM) proposed by Friedman & Stuetzle (1981). The GAM projects higher-dimensional features into lower-dimensional space to apply smoothing functions to build a restricted class of non-parametric regression models. GAM is less affected by the curse of dimensionality compared to directly using smoothing in a higher-dimensional space. For a given set of processes $J \subseteq [D] = \{1, \ldots, D\}$, the traditional GAM using one-dimensional projections is defined as $\log \lambda_J(\mathbf{t}) = \sum_{j \in J} f_j(t^{(j)}) - \beta_J$ with some smoothing function $f_j$.

In this paper, we extend it to include higher-order interactions between features in GAM by introducing terms that represent the multiplicative product between events. The *k-th order GAM* is defined as

$$\log \lambda_J(\mathbf{t}) = \sum_{j \in J} f_{\{j\}}(t^{(j)}) + \sum_{j_1, j_2 \in J} f_{\{j_1, j_2\}}(t^{(j_1)}, t^{(j_2)}) + \cdots + \sum_{j_1, \ldots, j_k \in J} f_{\{j_1, \ldots, j_k\}}(t^{(j_1)}, \ldots, t^{(j_k)}) - \beta_J$$
$$= \sum_{I \subseteq J, |I| \leq k} f_I(\mathbf{t}^{(I)}) - \beta_J, \tag{4}$$

where $\mathbf{t}^{(I)} \in \mathbb{R}^{|I|}$ denotes the subvector $(\mathbf{t}^{(j)})_{j \in I}$ of $\mathbf{t}$ with respect to $I \subseteq [D]$. The function $f_I : \mathbb{R}^{|I|} \to \mathbb{R}$ is a smoothing function to fit the data, and the normalization constant $\beta_J$ for the intensity function is obtained as $\beta_J = \int \lambda_J(\mathbf{t})dt = \int \exp(\sum_{I \subseteq J, |I| \leq k} f_I(\mathbf{t}^{(I)}))dt$. The definition of the additive model is in the same form as Equation (3). In particular, if we compare Equation (3) and Equation (4), we can see that the smoothing function $f$ in Equation (3) is realized as the summation over lower-dimensional projections in Equation (4).

Learning of a continuous function using lower-dimensional projections is well known because of the *Kolmogorov-Arnold representation theorem*, which states that:

**Theorem 1** (Kolmogorov–Arnold Representation Theorem (Braun & Griebel, 2009; Kolmogorov, 1957)). *Any multivariate continuous function can be represented as a superposition of one–dimensional functions; that is,* $f(t_1, \ldots, t_n) = \sum_{q=1}^{2n+1} f_q\left(\sum_{p=1}^n g_{q,p}(t_p)\right).$

Braun (2009) showed that the GAM is an approximation to the general form presented in Kolmogorov-Arnold representation theorem by replacing the range $q \in \{1, \ldots, 2n+1\}$ with $I \subseteq J$ and the inner function $g_{q,p}$ by the identity if $q = p$ and zero otherwise, yielding $f(\mathbf{t}) = \sum_{I \subseteq J} f_I(\mathbf{t}^{(I)})$.

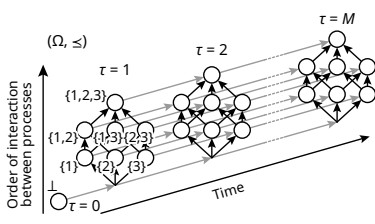

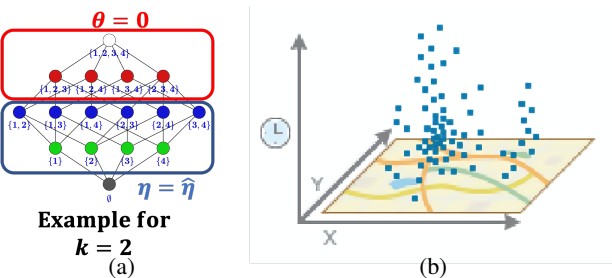

Figure 1: Partial order structured sample space $(\Omega, \preceq)$ with $D = 3$. Each node represents a state and and the directed edge represents the direction of the partial ordering.

Figure 2: (a) A visualisation of the truncated parameter space to approximate the joint intensity function with $D = 4$ and $k = 2$. (b) A visualization of the input datasets, where the blue points represent events with two spatial dimensions and one time dimension.

Interestingly, the canonical form for additive models in Equation (4) can be rearranged to be in the same form as Kolmogorov-Arnold representation theorem. By letting $f(\mathbf{t}) = \sum_{I \subseteq J} f_I(\mathbf{t}^{(I)}) = g^{-1}(\lambda(\mathbf{t}))$ and $g(\cdot) = \exp(\cdot)$, we have

$$\lambda_J(\mathbf{t}) = \frac{1}{\exp(\beta_J)} \exp\left(\sum_{I \subseteq J} f_I\left(\mathbf{t}^{(I)}\right)\right) \propto \exp\left(\sum_{I \subseteq J} f_I\left(\mathbf{t}^{(I)}\right)\right), \tag{5}$$

where we assume $f_I(\mathbf{t}^{(I)}) = 0$ if $|I| > k$ for the $k$-th order model and $1/\exp(\beta_J)$ is the normalization term for the intensity function. Based on the Kolmogorov-Arnold representation theorem, generalized additive models are able to learn the intensity of the higher-order interaction between Poisson processes by using projections into lower dimensional spaces. The log-likelihood function for a $k$th-order model is obtained by substituting $\lambda_J(\mathbf{t})$ in Equation (4) into $\lambda(\mathbf{t})$ in Equation (3),

$$\log p\left(\{\mathbf{t}\}_{i=1}^N | \lambda(\mathbf{t})\right) = \sum_{i=1}^{N} \sum_{I \subseteq J, |I| \leq k} f_I\left(\mathbf{t}^{(I)}\right) - \beta', \tag{6}$$

where $\beta'$ is a constant given by $\beta' = \int \lambda(\mathbf{t}) d\mathbf{t} + \sum_{I \subseteq J} \beta_J$. In the following subsection we introduce a log-linear formulation equipped with partially ordered sample space, which aligns with the GAM formulation in Equation (5).

## 2.2 ADDITIVE POISSON PROCESS

We introduce our key technical contribution in this section. We introduce a log-linear formulation called the *additive Poisson process* to estimate the parameters for the higher-order interactions in equation 6. We begin by discretizing the time window $[0, T]$ into $M$ bins and treat each bin as a natural number $\tau \in [M] = \{1, 2, \ldots, M\}$ for each process. The discretization avoids the need to compute the intractable integral in the likelihood function in Equation (6). The discretization approach tends to perform better in high-dimension compare to the alternative approaches such as variational inference. We assume that $M$ is predetermined by the user. First we introduce a structured space for the Poisson process to incorporate interactions between processes. Let $\Omega = \left\{(J, \tau) \mid J \in 2^{[D]} \setminus \emptyset, \tau \in [M]\right\} \cup \{(\perp, 0)\}$. We define the *partial order* $\preceq$ (Davey & Priestley, 2002) on $\Omega$ as

$$\omega = (J, \tau) \preceq \omega' = (J', \tau') \iff J \subseteq J' \text{ and } \tau \leq \tau', \qquad \text{for each } \omega, \; \omega' \in \Omega, \tag{7}$$

and $(\perp, 0) \preceq \omega$ for all $\omega \in \Omega$, which is illustrated in Figure 1. The relation $J \subseteq J'$ is used to model any-order interactions between Poisson processes (Luo & Sugiyama, 2019) (Amari, 2016, Section 6.8.4) and each $\tau$ in $(J, \tau)$ represents the auto-regressive component ("time") in our model. The symbol $\perp$ denotes the least element in the partial order structure, which is required to normalize probabilities in the log-linear model. Each node $\omega$ in the partially ordered set (poset) [1] represents the state of the sample space and the arrows in Figure 1 represent the partial order relationship between two nodes[2]; that is, if $\omega \rightarrow \omega'$, then $\omega \preceq \omega'$.

---

[1] In information geometry, this corresponds to *hypergraphs* which have *simplicial complex* structure Ay et al. (2018, Section 2.9).

[2] This graph structure should not be confused with the graph structure studied in graphical models, where the nodes typically represent a random variable and the arrows represent the relationship between the two random variables.

Intuitively, the greatest node for each $\tau \in [M]$, which is $(\{1, 2, 3\}, \tau)$ in Figure 1, represents the multi-dimensional Poisson process. Other nodes represent projections onto lower-dimensional space that correspond to the marginalized observations; for example, $\{\{1\}, \{2\}, \{3\}\}$ and $\{\{1, 2\}, \{1, 3\}, \{2, 3\}\}$ represent the first- and second-order processes. Using our example in Introduction, where we wanted to estimate the intensity function of a pick-up event of a taxi, $\{1\}$ and $\{2\}$ correspond to spacial coordinates $x$ and $y$, respectively, and $\{3\}$ to the day of the week $W$, and $\tau$ represents the (discretized) observation time. We can then update our belief to model the second-order intensity function using observations of the second order events. For example, $\{1, 2\}, \{1, 3\}, \{2, 3\}$ represents an event occurring at $\{x, y\}$, $\{x, W\}$, and $\{y, W\}$. We can then continue this process to an arbitrary order of interac-

**Algorithm 1** Additive Poisson Process (APP)

1: **Function** APP($\{\mathbf{t}_i\}_{i=1}^N$, $\mathcal{S}$, $M$, $h$):
2: Initialize $\Omega$ with the number $M$ of bins
3: Apply Gaussian Kernel with bandwidth $h$ on $\{\mathbf{t}_i\}_{i=1}^N$ to compute $\hat{p}$
4: Compute $\hat{\boldsymbol{\eta}} = (\hat{\eta}_s)_{s \in \mathcal{S}}$ from $\hat{p}$
5: Initialize $\boldsymbol{\theta} = (\theta_s)_{s \in \mathcal{S}}$ (randomly or $\theta_s = 0$)
6: **repeat**
7:     Compute $p$ using the current $\boldsymbol{\theta} = (\theta_s)_{s \in \mathcal{S}}$
8:     Compute $\boldsymbol{\eta} = (\eta_s)_{s \in \mathcal{S}}$ from $p$
9:     $\Delta \boldsymbol{\eta} \leftarrow \boldsymbol{\eta} - \hat{\boldsymbol{\eta}}$
10:     Compute the Fisher information matrix $\mathbf{G}$ using Equation (11)
11:     $\boldsymbol{\theta} \leftarrow \boldsymbol{\theta} - \mathbf{G}^{-1}\Delta\boldsymbol{\eta}$
12: **until** convergence of $\boldsymbol{\theta} = (\theta_s)_{s \in \mathcal{S}}$
13: **End Function**

tions. Later on in this section we introduce the mathematics to estimate the higher-order function using a restricted number of lower-dimensional projections.

The domain of $\tau$ can be generalized from $[M]$ to $[M]^D$ to take different time stamps into account, while in the following we assume that observed time stamps are always the same across processes for simplicity. Our experiments in the next section demonstrate that we can still accurately estimate the density of processes. Our model can be applied not only to time-series data, but to any sequential data.

On any set equipped with a partial order, we can introduce a *log-linear model* (Sugiyama et al., 2017). Let us assume that a parameter domain $\mathcal{S} \subseteq \Omega$ is given. For a partially ordered set $(\Omega, \preceq)$, the log-linear model with parameters $(\theta_s)_{s \in \mathcal{S}}$ is introduced as

$$\log p(\omega; \theta) = \sum_{s \in \mathcal{S}} \mathbf{1}_{[s \preceq \omega]} \theta_s - \psi(\theta) \tag{8}$$

for each $\omega \in \Omega$, where $\mathbf{1}_{[\cdot]} = 1$ if the statement in $[\cdot]$ is true and 0 otherwise, and $\psi(\theta) \in \mathbb{R}$ is the partition function uniquely obtained as $\psi(\theta) = \log \sum_{\omega \in \Omega} \exp(\sum_{s \in \mathcal{S}} \mathbf{1}_{[s \preceq \omega]} \theta_s) = -\theta_{(\perp, 0)}$. A special case of this formulation coincides with the density function of the *Boltzmann machines* (Sugiyama et al., 2018; Luo & Sugiyama, 2019).

Here there is a clear correspondence between the log-linear formulation and that in the form of Kolmogorov-Arnold representation theorem in Equation (5) if we rewrite Equation (8) as

$$p(\omega; \theta) = \frac{1}{\exp \psi(\theta)} \exp\left(\sum_{s \in \mathcal{S}} \mathbf{1}_{[s \preceq \omega]} \theta_s\right) \propto \exp\left(\sum_{s \in \mathcal{S}} \mathbf{1}_{[s \preceq \omega]} \theta_s\right). \tag{9}$$

We call this model with $(\Omega, \preceq)$ defined in Equation (7) the additive Poisson process, which represents the intensity $\lambda$ as the joint distribution across all possible states. The intensity $\lambda$ of the multi-dimensional Poisson process given via the GAM in Equation (5) is fully modeled (parameterized) by Equation (8) and each intensity $f_I(\cdot)$ is obtained as $\theta_{(I, \cdot)}$. To consider the $k$-th order model, we consistently use the parameter domain $\mathcal{S}$, given as $\mathcal{S} = \{(J, \tau) \in \Omega \mid |J| \leq k\}$, where $k$ is an input parameter to the model that specifies the upper bound of the order of interactions. This means that $\theta_s = 0$ for all $s \notin \mathcal{S}$. Note that our model is well-defined for any subset $\mathcal{S} \subseteq \Omega$ and the user can use an arbitrary domain in applications. A visualization of the truncated parameter space is shown in Figure 2a.

For a given $J \subseteq [D]$ and each bin $\tau$ with $\omega = (J, \tau)$, the empirical probability $\hat{p}(\omega)$ of input observations is given as

$$\hat{p}(\omega) = \frac{1}{Z} \sum_{I \subseteq J} \sigma_I(\boldsymbol{\tau}), \qquad Z = \sum_{\omega \in \Omega} \hat{p}(\omega), \qquad \sigma_I(\boldsymbol{\tau}) := \frac{1}{N h_I} \sum_{i=1}^N K\left(\frac{\boldsymbol{\tau}^{(I)} - \mathbf{t}_i^{(I)}}{h_I}\right) \tag{10}$$

for each discretized state $\omega = (J, \tau)$, where $\boldsymbol{\tau} = (\tau, \dots, \tau) \in \mathbb{R}^D$. The function $\sigma_I$ performs smoothing on time stamps $\mathbf{t}_1, \dots, \mathbf{t}_N$, which is the kernel smoother proposed by Buja et al. (1989).

The function $K$ is a kernel and $h_I$ is the bandwidth for each projection $I \subseteq [D]$. We use the Gaussian kernel as $K$ to ensure that probability is always nonzero, meaning that the definition of the kernel smoother coincides with the kernel estimator of the intensity function proposed by Schäbe (1993).

## 2.3 OPTIMIZATION

Given an empirical distribution $\hat{p}$ defined in Equation (10), the task is to learn the parameter $(\theta_s)_{s \in \mathcal{S}}$ such that the distribution via the log-linear model in Equation (8) is as close to $\hat{p}$ as much as possible. Let us define $\mathfrak{S}_\mathcal{S} = \{p \mid \theta_s = 0 \text{ if } s \notin \mathcal{S}\}$, which is the set of distributions that can be represented by the log-linear model using the parameter domain $\mathcal{S}$. Then the objective function is given as $\min_{p \in \mathfrak{S}_\mathcal{S}} D_{\mathrm{KL}}(\hat{p}, p)$, where $D_{\mathrm{KL}}(\hat{p}, p) = \sum_{\omega \in \Omega} \hat{p} \log(\hat{p}/p)$ is the KL divergence from $\hat{p}$ to $p$. In this optimization, let $p^*$ be the learned distribution from the sample with infinitely large sample size and let $p$ be the learned distribution for each sample. Then we can lower bound the uncertainty (variance) $\mathbb{E}[D_{\mathrm{KL}}(p^*, p)]$ by $|\mathcal{S}|/2N$ (Barron & Hengartner, 1998).

Thanks to the well-developed theory of *information geometry* (Amari, 2016) for the log-linear model (Amari, 2001), it is known that this problem can be solved by *e-projection*, which coincides with the maximum likelihood estimation and is always *convex optimization* (Amari, 2016, Chapter 2.8.3). The gradient with respect to each parameter $\theta_s$ is obtained by $(\partial/\partial \theta_s) D_{\mathrm{KL}}(\hat{p}, p) = \eta_s - \hat{\eta}_s$, where $\eta_s = \sum_{\omega \in \Omega} \mathbf{1}_{[\omega \succeq s]} p(\omega)$. The value $\eta_s$ is known as the expectation parameter (Sugiyama et al., 2017) and $\hat{\eta}_s$ is obtained by replacing $p$ with $\hat{p}$ in the above equation. If $\hat{\eta}_s = 0$ for some $s \in \mathcal{S}$, we remove $s$ from $\mathcal{S}$ to ensure that the model is well-defined.

Let $\mathcal{S} = \{s_1, \dots, s_{|\mathcal{S}|}\}$ and $\boldsymbol{\theta} = [\theta_{s_1}, \dots, \theta_{s_{|\mathcal{S}|}}]^T$, $\boldsymbol{\eta} = [\eta_{s_1}, \dots, \eta_{s_{|\mathcal{S}|}}]^T$. We can always use the *natural gradient* (Amari, 1998) as the closed form solution of the Fisher information matrix is always available (Sugiyama et al., 2017). The update step is $\boldsymbol{\theta}_{\mathrm{next}} = \boldsymbol{\theta} - \mathbf{G}^{-1}(\boldsymbol{\eta} - \hat{\boldsymbol{\eta}})$, where the Fisher information matrix $\mathbf{G}$ is obtained as

$$g_{ij} = \frac{\partial}{\partial \theta_{s_i} \partial \theta_{s_j}} D_{\mathrm{KL}}(\hat{p}, p) = \sum_{\omega \in \Omega} \mathbf{1}_{[\omega \succeq s_i]} \mathbf{1}_{[\omega \succeq s_j]} p(\omega) - \eta_{s_i} \eta_{s_j}. \tag{11}$$

Theoretically the Fisher information matrix is numerically stable to perform a matrix inversion. However, computationally, floating point errors may cause the matrix to become indefinite. To overcome this issue, a small positive value is added along the main diagonal of the matrix. This technique is known as jitter and it is used in areas like Gaussian processes to ensure that the covariance matrix is computationally positive semi-definite (Neal, 1999).

The pseudocode for APP is shown in Algorithm 1. The time complexity of computing line 7 is $\mathcal{O}(|\Omega||\mathcal{S}|)$. This means when implementing the model using gradient descent, the time complexity of the model is $\mathcal{O}(|\Omega||\mathcal{S}|^2)$ to update the parameters in $\mathcal{S}$ for each iteration. For natural gradient, the cost of inverting the Fisher information matrix $G$ is $\mathcal{O}(|\mathcal{S}|^3)$; therefore, the time complexity to update the parameters in $\mathcal{S}$ is $\mathcal{O}(|\mathcal{S}|^3 + |\Omega||\mathcal{S}|)$ for each iteration. The time complexity for natural gradient is significantly higher higher because of the requirement to invert the fisher information matrix; if the number of parameters is small, it is more efficient to use natural gradient because it requires significantly fewer iterations. However, if the number of parameters is large, it is more efficient to use gradient descent.

## 3 EXPERIMENTS

We perform experiments using two-dimensional synthetic data, higher-dimensional synthetic data, and real-world data to evaluate the performance of our proposed approach. Our code is implemented in Python 3.7.5 with NumPy version 1.8.2 and the experiments are run on Ubuntu 18.04 LTS with an Intel i7-8700 6c/12t with 16GB of memory [3]. In experiments with synthetic data, we simulate random events using Equation (1). We generate an intensity function using a mixture of Gaussians, where the mean is drawn from a uniform distribution and the covariance is drawn from an inverted Wishart distribution. The intensity function is then the density function multiplied by the sample size. The synthetic data is generated by directly drawing a sample from the probability density function . An arbitrary number of samples is drawn from the mixture of Gaussians. We

---

[3]The code is available in the supplementary material and will be publicly available online.

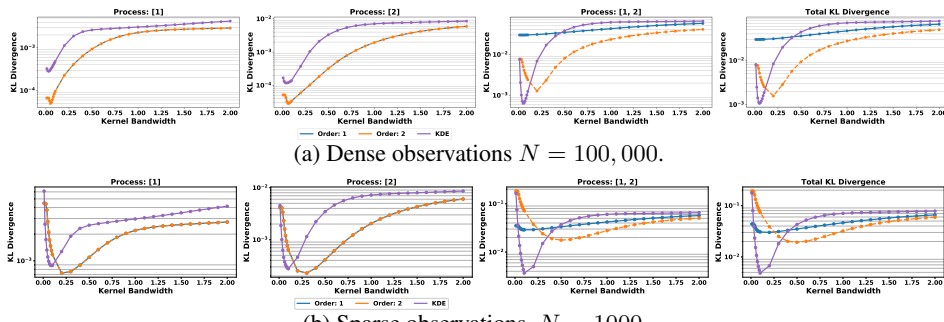

(a) Dense observations $N = 100,000$.

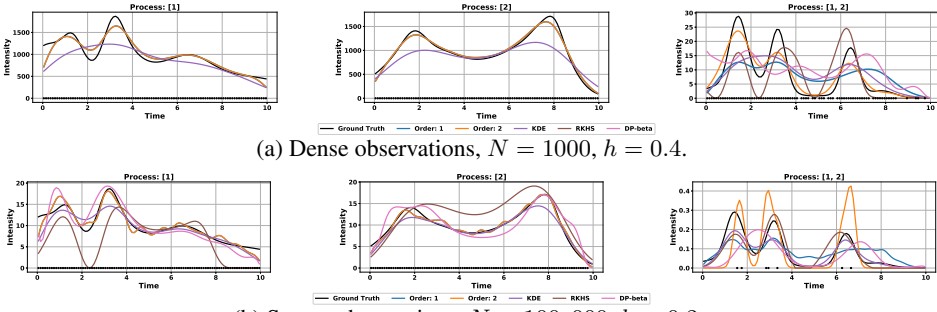

(b) Sparse observations, $N = 1000$.

Figure 3: KL Divergence for second-order Poisson process. The order of the model (color of the line) represents the $k$-th order model, i.e., $k = 1$ (blue) and $k = 2$ (orange).

(a) Dense observations, $N = 1000$, $h = 0.4$.

(b) Sparse observations, $N = 100,000$, $h = 0.2$.

Figure 4: Intensity function of two dimensional processes. Dots represent observations. Left: Represents marginalized observation of the first dimension. Middle: Represents marginalized observation of the second dimension. Right: The joint observation of dimension 1 and 2. The order of the model (color of the line) represents the $k$-th order model, i.e., $k = 1$ (blue) and $k = 2$ (orange).

Table 1: The lowest KL divergence from the ground truth distribution to the obtained distribution on two types of single processes ([1] and [2]) and joint process of them ([1,2]). APP-# represents the order of the Additive Poisson Process. Missing values mean that the computation did not finish within two days.

|        | Process | APP-1   | APP-2   | KDE     | RKHS   | DP-beta |
|--------|---------|---------|---------|---------|--------|---------|
| Dense  | [1]     | **4.98e-5** | **4.98e-5** | 2.81e-4 | -      | -       |
|        | [2]     | **2.83e-5** | **2.83e-5** | 1.17e-4 | -      | -       |
|        | [1,2]   | 2.98e-2 | 1.27e-3 | **6.33e-4** | 4.09e-2 | 4.54e-2 |
| Sparse | [1]     | **7.26e-4** | **7.26e-4** | 8.83e-4 | 1.96e-2 | 2.62e-3 |
|        | [2]     | **2.28e-4** | **2.28e-4** | 2.76e-4 | 2.35e-3 | 2.49e-3 |
|        | [1,2]   | 2.88e-2 | 1.77e-2 | **3.67e-3** | 1.84e-2 | 3.68e-2 |

Table 2: Negative test log-likelihood for the New York Taxi data. Single processes ([T] and [W]) and joint process of them ([T,W]). APP-# represents the order of the Additive Poisson Process.

|     | Process | APP-1  | APP-2  | KDE    | RKHS   | DP-beta |
|-----|---------|--------|--------|--------|--------|---------|
| Jan | [T]     | 714.07 | 714.07 | **713.77** | 728.13 | 731.01  |
|     | [W]     | 745.60 | 745.60 | **745.23** | 853.42 | 790.04  |
|     | [T,W]   | 249.60 | **246.05** | 380.22 | 259.29 | 260.30  |
| Feb | [T]     | **713.43** | **713.43** | 755.71 | 795.61 | 765.76  |
|     | [W]     | **738.66** | **738.66** | 773.65 | 811.34 | 792.10  |
|     | [T,W]   | 328.84 | **244.21** | 307.86 | 334.31 | 326.52  |
| Mar | [T]     | **716.72** | **716.72** | 733.74 | 755.48 | 741.28  |
|     | [W]     | **738.06** | **738.06** | 816.99 | 853.33 | 832.43  |
|     | [T,W]   | 291.20 | **246.19** | 289.69 | 328.47 | 300.36  |

then run our models and compare with Kernel Density Estimation (KDE) (Rosenblatt, 1956), an inhomogeneous Poisson process whose intensity is estimated by a reproducing kernel Hilbert space formulation (RKHS) (Flaxman et al., 2017), and a Dirichlet process mixture of Beta distributions (DP-beta) (Kottas, 2006; Kottas & Sansó, 2007). The hyper-parameters $M$ and $h$ in our proposed model are selected using grid search and cross-validation. For situations where a validation set is not available, then $h$ could be selected using a rule of thumb approach such as Scott's Rule (Scott, 2015) and $M$ could be selected empirically from the input data by computing the time interval of the joint observation.

## 3.1 EXPERIMENTS ON TWO-DIMENSIONAL PROCESSES

For our experiment, we use 20 Gaussian components and simulate a dense case with 100,000 observations and a sparse case with 1,000 observations within the time frame of 10 seconds. We consider that a joint event occurs if the two events occur 0.1 seconds apart. Figure 3a and Figure 3b compare the KL divergence between the first- and second-order models and plots in Figure 4 are the corresponding intensity functions. In the first-order processes, both first- and second-order models

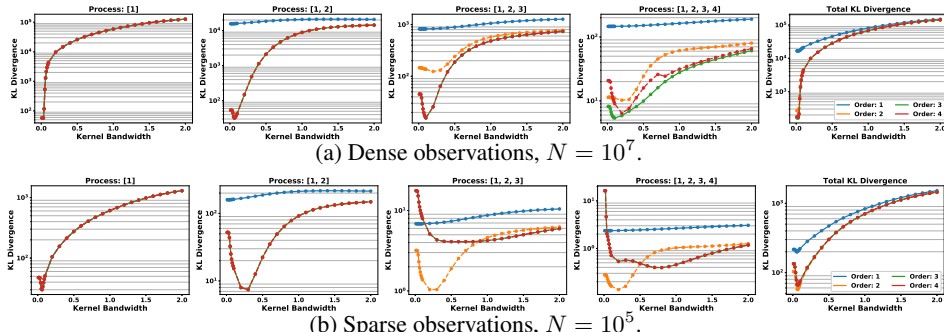

(a) Dense observations, $N = 10^7$.

(b) Sparse observations, $N = 10^5$.

Figure 5: KL Divergence for fourth-order Poisson process. We selected four representative examples for our experimental results, full results available in the supplementary material. The line color signifies the order of the model, i.e., $k = 1$ (blue), $k = 2$ (orange), $k = 3$ (green) and $k = 4$ (red).

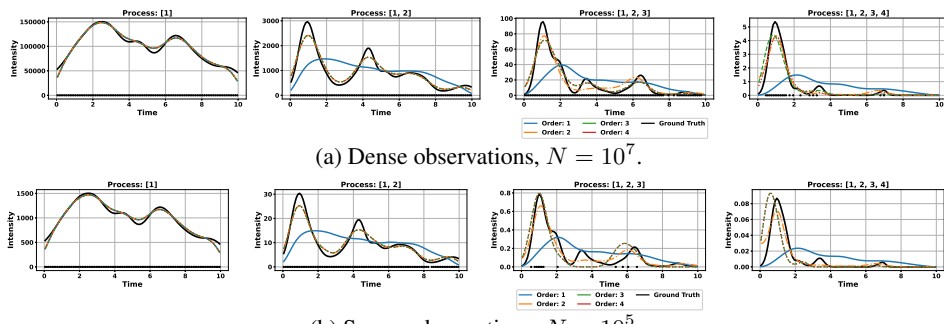

(a) Dense observations, $N = 10^7$.

(b) Sparse observations, $N = 10^5$.

Figure 6: Intensity function of higher dimensional processes. Dots represent observations. We have selected four representative examples for our experimental results, full results available in the supplementary material. The order of the model (color of the line) represents the $k$-th order model, i.e., $k = 1$ (blue), $k = 2$ (orange), $k = 3$ (green) and $k = 4$ (red).

have the same performance. This is expected, as both of the models can treat first-order interactions and are able to learn the empirical intensity function exactly, which is the superposition of the one-dimensional projection of the Gaussian kernels on each observation. For the second-order process, the second-order model performs better than the first-order model because it is able to directly learn the intensity function from the projection onto the two-dimensional space. In contrast, the first-order model must approximate the second-order process using the observations from the first-order processes. In the sparse case, the second-order model performs better when the correct bandwidth is selected.

Table 1 compares our approach APP with other state-of-the-art approaches. APP performs the best for first-order processes in both the sparse and dense experiments. Experiments for RKHS and DP-beta were unable to complete running within two days for the dense experiment. In the second-order process, our approach was outperformed by KDE, while the second-order APP was able to outperform both RKHS and DP-beta process for both sparse and dense experiments. Figures 3a and 3b show that KDE is sensitive to changes in bandwidth, which means that, for any practical implementation of the model, second-order APP with a less sensitive bandwidth is more likely to learn a more accurate intensity function when the ground truth is unknown.

## 3.2 EXPERIMENTS ON HIGHER-DIMENSIONAL PROCESSES

We generate a fourth-order process to simulate the behavior of the model in higher dimensions. The model is generalizable to higher dimensions, but it is difficult to demonstrate results for processes higher than fourth order. For our experiment, we generate an intensity function using 50 Gaussian components and draw a sample with the size of $10^7$ for the dense case and that with the size of $10^5$ for the sparse case. We consider the joint event to be the time frame of 0.1 seconds.

We were not able to run comparison experiments with other models because they are unable to learn when there are no or few joint observations in third- and fourth-order processes. In addition, the time complexity is too high to learn from joint observations in first- and second-order processes because

all the other models have their time complexity proportional to the number of observations. The time complexity for KDE is $\mathcal{O}(N^D)$ for the dimensionality with $D$, while DP-beta is $\mathcal{O}(N^2K)$, where $K$ is the number of clusters, and RKHS is $\mathcal{O}(N^2)$ for each iteration with respect to the sample size $N$, where DP-beta and RKHS are applied directly on the joint observation as they cannot use the projections in lower-dimensional space. KDE is able to make an estimation of the intensity function using projections in lower-dimensional space, but it was too computationally expensive to complete running the experiment. By contrast, our model is more efficient because the time complexity is proportional to the number of bins in our model. The time complexity of APP for each iteration is $\mathcal{O}(|\Omega||\mathcal{S}|)$, where $|\Omega| = M^D$ and $|\mathcal{S}| = \sum_{c=1}^{k} \binom{D}{c}$. Our model scales combinatorially with respect to the number of dimensions. However, this is unavoidable for any model that directly takes into account the high-order interactions. For practical applications, the number of dimensions $D$ and the order of the model $k$ is often small, making it feasible to compute.

In Figure 5a, we observe similar behavior in the model, where the first-order processes fit precisely to the empirical distribution generated by the Gaussian kernels. The third-order model is able to predict better on the fourth-order process. This is because the observation shown in Figure 6a is largely sparse and learning from the observations directly may overfit. A lower-dimensional approximation is able to provide a better result in the third-order model. Similar trends can be seen in the sparse case, as shown in Figure 5b, where a second-order model is able to produce better estimation in third- and fourth-order processes. The observations are extremely sparse, as seen in Figure 6b, where there are only a few observations or no observations at all to learn the intensity function.

### 3.3 Uncovering Common Patterns in the New York Taxi Dataset

We demonstrate the capability of our model on the 2016 Green Taxi Trip dataset[4], which is a open source dataset with a CC0: Public Domain licences. We are interested in finding the common pick-up patterns across Tuesdays and Wednesdays. We define a common pick-up time to be within 1-minute intervals of each other between the two days. We have chosen to learn an intensity function using the Poisson process for Tuesday and Wednesday and a joint process for both of them. The joint process uncovers the common pick-up patterns between the two days. We have selected to use the first two Tuesdays and Wednesdays in January 2016 as our training and validation sets and the Tuesday and Wednesday of the third week of January 2016 as our testing set. We repeat the same experiment for February and March.

We show our results in Table 2, where we use the negative test log-likelihood as an evaluation measure. APP-2 has consistently outperformed all the other approaches for the joint process between Tuesday and Wednesday. In addition, for the individual process, APP-1 and -2 also showed the best result for February and March. These results demonstrate the effectiveness of our model in capturing higher-order interactions between processes, which is difficult for the other existing approaches.

## 4 Conclusion

We have proposed a novel framework, called *Additive Poisson Process* (APP), to learn the intensity function of the higher-order interaction between Poisson processes using observations projected into lower-dimensional spaces. We formulated our proposed model using the *log-linear model* and optimize it using information geometric structure of the distribution space. We drew parallels between our proposed model and *generalized additive model* and showed the ability to learn from lower dimensional projections via the *Kolmogorov-Arnold representation theorem*. Our empirical results show the superiority of our method when learning the higher-order interactions between Poisson processes and when there are no or extremely sparse joint observations. Our model is also robust to varying sample sizes. Our approach provides a novel formulation to learn the joint intensity function which typically has extremely low intensity. There is enormous potential to apply APP to real-world applications, where higher order interaction effects need to be modeled such as transportation, finance and ecology.

---

[4]https://data.cityofnewyork.us/Transportation/2016-Green-Taxi-Trip-Data/hvrh-b6nb

ETHICS STATEMENT

This paper is intended to be written in a neutral tone. Any suggestion of any application that may be considered as unethical or have a negative social impact is only coincidental and not our intention. The results of this work should only be used to have a positive impact in society.

REPRODUCIBILITY STATEMENT

Our model is implemented in Python 3.7.5 with NumPy version 1.8.2. The code to reproduce our experiments is available in the supplementary material and will be made publicly available on GitHub after the paper is accepted.

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

APPENDIX

# A    RELATED WORK

## A.1    BAYESIAN INFERENCE FOR POISSON PROCESS

Learning an intensity function from sparse high-dimensional datasets is a challenging problem. It often comes a trade-off between computational complexity and numerical accuracy. For applications which are sparse with higher dimensionality, numerical accuracy is often sacrificed to approximate the intensity function.

The naïve approach using MCMC has cubic complexity $\mathcal{O}(n^3)$ for each dimension and increases exponentially with respect with additional input dimension, that is, $\mathcal{O}((n^3)^d)$, where $n$ refers to the number of observations and $d$ is the number of dimensions. One example of this approach is using a Dirichlet process mixture of Beta distribution as a prior for the intensity function of the Poisson process (Kottas, 2006; Kottas & Sansó, 2007). The solution from MCMC is often accurate and asymtopically converges to the true posterior intensity function. However, due to its computational complexity, it is infeasible to estimate any high-dimensional intensity function.

More recent techniques have attempted to scale up these approaches by using Gaussian processes as a functional prior to the intensity function (Samo & Roberts, 2015). However, the model complexity is $\mathcal{O}(n^2 k)$ for each dimension and is exponential with respect to the number of input dimensions, which is still infeasible to estimate any high-dimensional intensity function.

Variational inference (Lloyd et al., 2015) approaches that can be used to make Bayesian inference of Poisson process much more efficient scaling it up to the linear complexity $\mathcal{O}(n)$ for each dimension. However variational inference is not guaranteed to asymptotically converge to the true posterior distribution.

Our approach uses the discretization approach to scale up the model to higher dimensions. We use a graph (partial order) structure to allow the flexibility for domain expertise to specific how each dimension is treated and which interaction effects should be included into the model.

## A.2    POISSON FACTORIZATION

Our work is closely related to Poisson Factorization (Chi & Kolda, 2012), where random variables in a tensor are represented with a Poisson distributed or Poisson process likelihood. The tensor is usually used to represent some high-dimensional dataset such as contingency tables or other collection of counting datasets, which are often large and sparse. The objective of Poison Factorization is to decompose the high-dimensional sparse matrices into lower dimensional space, where we can find some meaningful latent structure.

The effectiveness of Poisson factorization for high dimensional datasets makes it ideal to analyze spatial-temporal problems consisting of sparse count data. One example of this work is Bayesian Poisson Tucker decomposition (BPTD)  (Schein et al., 2016), where a dataset of interaction events is represented as a set of $N$ events, each of which consists of a pair of a token $\mathbf{e}_i$ that encodes certain features and time, that is, $(\mathbf{e}_i, \mathbf{t}_i)$. BPTD uses an MCMC inference algorithm to learn the latent structure, which is based on an extension of stochastic block models (SBM) (Nowicki & Snijders, 2001) with a Poisson likelihood.

Our approach provides a generalization of this idea of Poisson Factorization by using Legendre tensor decomposition (Sugiyama et al., 2018) and demonstrating its ability on a spatial-temporal problem. Our optimization is much more efficient as it is guided by gradients to minimize the KL-divergence. Our approach also contains a graph structure which allows domain experts to encode certain properties into the model.

---

**Algorithm 2** Thinning Algorithm for non-homogenous Poisson Process

---

1: **Function** Thinning Algorithm $(\lambda(t), T)$:
2: $n = m = 0, t_0 = s_0 = 0, \bar{\lambda} = \sup_{0 \leq t \leq T} \lambda(t)$
3: **repeat**
4:    $u \sim \text{uniform}(0, 1)$
5:    $w = -\frac{1}{\bar{\lambda}} \ln u \ \{w \sim \text{exponential}(\bar{\lambda})\}$
6:    $s_{m+1} = s_m + w$
7:    $D \sim \text{uniform}(0, 1)$
8:    **if** $D \leq \frac{\lambda(s_{m+1})}{\bar{\lambda}}$ **then**
9:      $t_{n+1} = s_{m+1}$
10:     $n = n + 1$
11:   **else**
12:     $m = m + 1$
13:   **end if**
14:   **if** $t_n \leq T$ **then**
15:     **return** $\{t_k\}_{k=1,2,\ldots,n}$
16:   **else**
17:     **return** $\{t_k\}_{k=1,2,\ldots,n-1}$
18:   **end if**
19: **until** $s_m \leq T$
20: **End Function**

---

# B ADDITIONAL EXPERIMENTS

## B.1 BANDWIDTH SENSITIVITY ANALYSIS

Our first experiment is to demonstrate the ability for our proposed model to learn an intensity function from samples. We generate a Bernoulli process with probably of $p = 0.1$ to generate samples for every 1 seconds for 100 seconds to create a toy problem for our model. This experiment is to observe the behaviour of varying the bandwidth in our model. In Figure 7a, we observe that applying no kernel, we learn the deltas of each individual observation. When we apply a Gaussian kernel, the output of the model for the intensity function is much more smooth. Increasing the bandwidth of the kernel will provide a wider and much smoother function. Between the 60 seconds and 80 seconds mark, it can be seen when two observations have overlapping kernels, the intensity function becomes larger in magnitude.

## B.2 ONE DIMENSIONAL POISSON PROCESS

A one dimensional experiment is simulated using Ogata's thinning algorithm (Ogata, 1981). We generate two experiments use the standard sinusoidal benchmark intensity function with a frequency of $20\pi$. The dense experiment has troughs with 0 intensity and peaks at 201 and the sparse experiment has troughs with 0 intensity and peaks at 2. Figure 7d shows the experimental results of the dense case, our model has no problem learning the intensity function. We compare our results using KL divergence between the underlying intensity function used to generate the samples to the intensity function generated by the model. Figure 7b shows that the optimal bandwidth is $h = 1$.

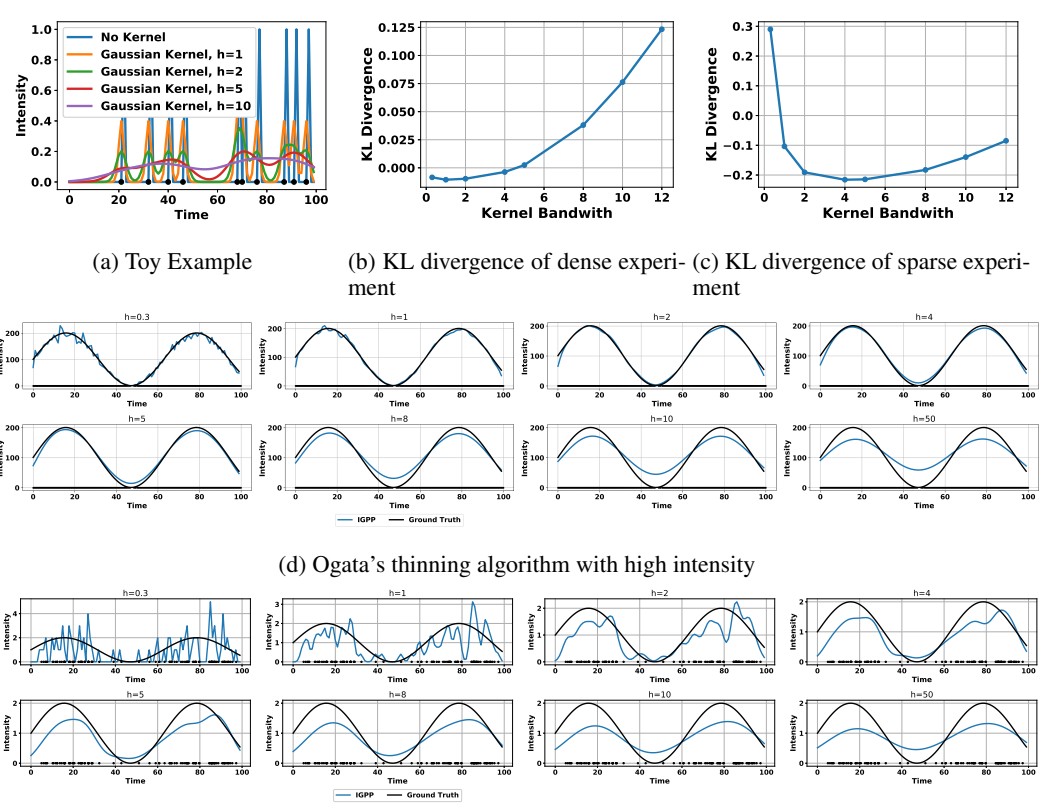

(a) Toy Example

(b) KL divergence of dense experiment

(c) KL divergence of sparse experiment

(d) Ogata's thinning algorithm with high intensity

(e) Ogata's thinning algorithm with low intensity

Figure 7: One dimensional experiments

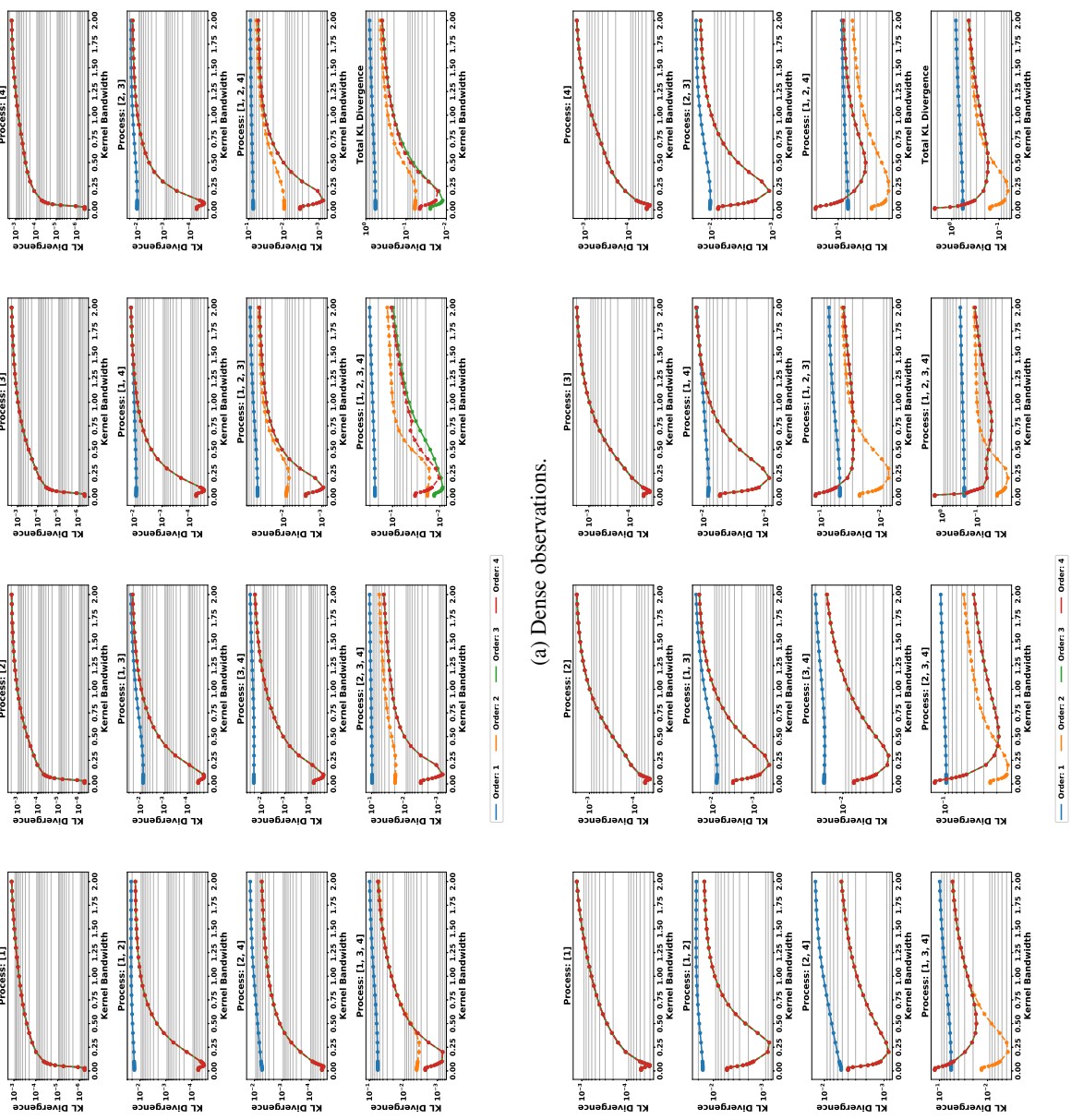

(a) Dense observations.

(b) Sparse observations.

Figure 8: KL Divergence for four-order Poisson process.

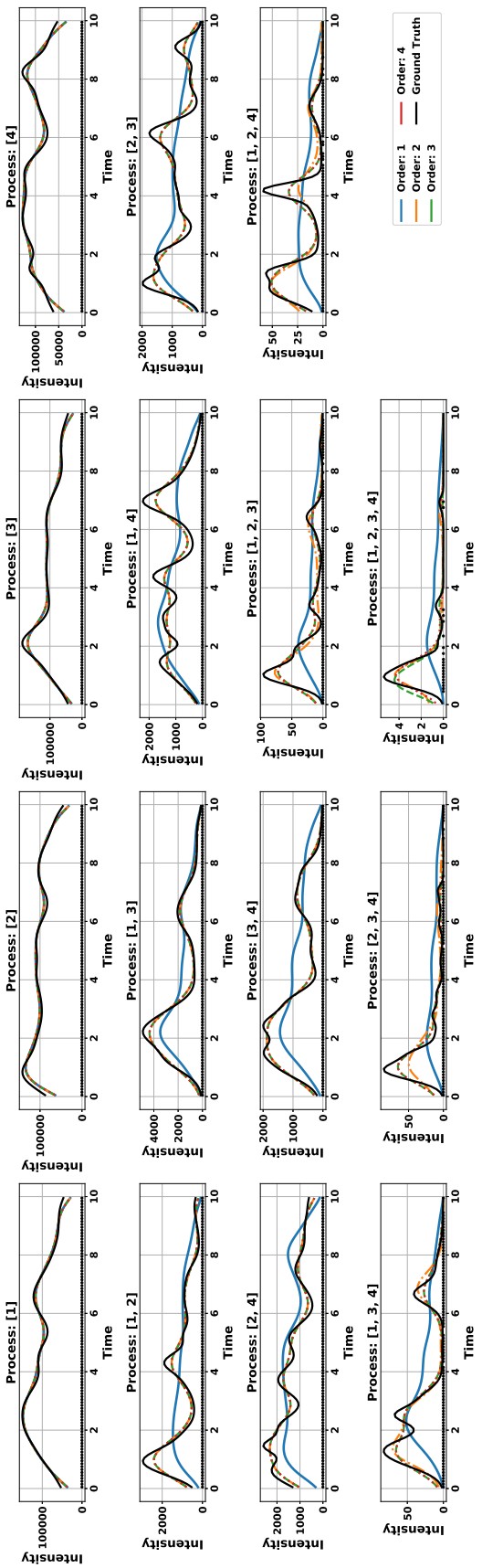

(a) Dense observations.

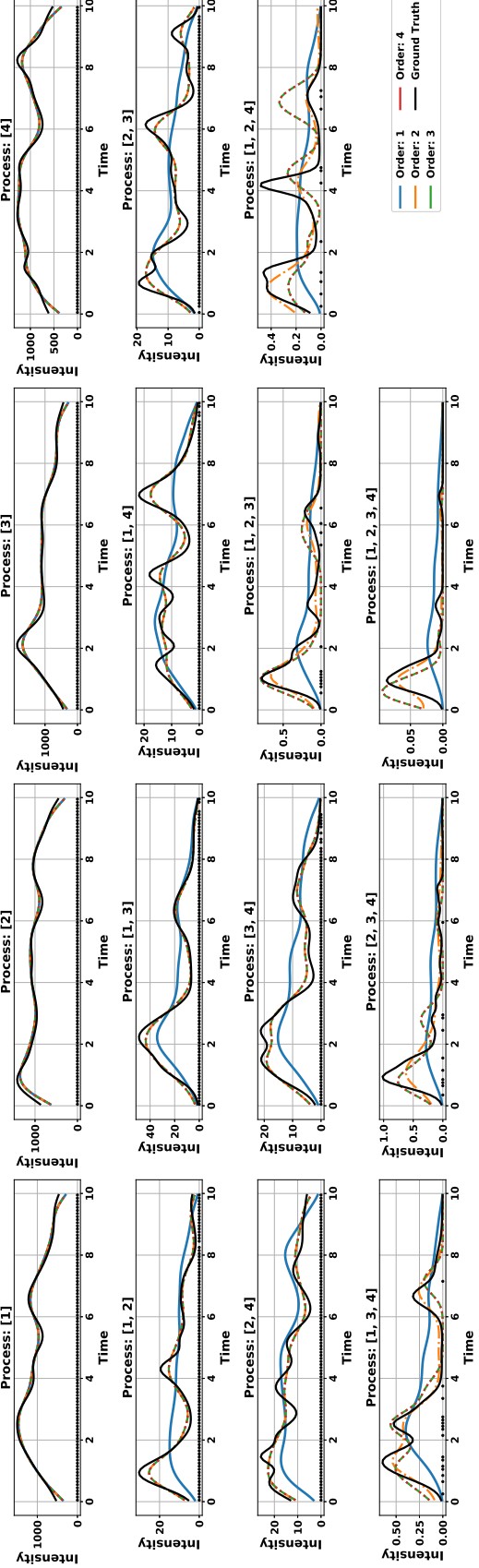

(a) Sparse observations.

Figure 10: Intensity function of higher dimensional processes. Dots represent observations.

