# OpenReview forum: "Additive Poisson Process: Learning Intensity of Higher-Order Interaction in Poisson Processes"
_ICLR.cc/2022/Conference — ICLR 2022 Submitted_

### Official Review · Reviewer_wbmT · 2021-10-19

**Correctness:** 2
**Technical Novelty And Significance:** 2
**Empirical Novelty And Significance:** 2
**Recommendation:** 3
**Confidence:** 4

**Main Review:**

This approach is interesting but it would deserve more works to further develop the idea. The paper is not motivated and clear enough which leaves the reader perplexed. Here are some remarks:


- Several statements in the paper are not well-justified by references, e.g. on KDE in the introduction.

- More details should be added on how GAM approximates the Kolmogorov-Arnold representation and how well it performs. For example, 'based on the K-A representation theorem, GAM are able to learn the intensity of the higher-order interaction between Poisson processes' It would be nice to add references and examples since it is the key of the paper.

- Motivations for introducing APP are not well explained. Background on Poisson Processes are not clear enough and do not give any insight about PP.

- The presentation of APP, in Section 2.2, should be clarified. It is not clear how to match the log-linear model (8) and the intensity defined in Sections 2 and 2.1, e.g. the taxi toy example at the top of page 5 makes things even more confusing.

- I would like to see experiments on the computation time of the approach compared to existing methods as it could be an advantage of this approach.

Minor comments:

- The notation \mathbf{t}_i in Section 2 is very confusing.

- Figure 2 is cited in the text before Figure 1.

- Section 2.2 The symbol $\bot$ should be defined before, or just after,  using it. Furthermore,  $\bot$ denotes the least element in the partial order structure but is used before 'partial order' definition. Also, I think $J$ belongs to $ 2^{|[D]|}$.

- Equation (5) is weird since you basically inverse log-function on equation (4) but the index $k$ disappears and appears again in (6).

- Figures are too small, it is difficult to see anything.

- "to arrive at": "to converge to"

**Summary Of The Paper:**

The paper propose to apply generalized additive models (GAM) to learn the intensity of the multi-dimensional Poisson processes (PP).

**Summary Of The Review:**

The paper is not motivated and clear enough. Furthermore, the proposed approach is empirically poorly studied while no theoretical guarantees are investigated. Several claims are not supported by references or proof.

---

> ### Author Response · Authors · 2021-11-18
> **Authors Response to Reviewer wbmT (1/2)**
>
> Thank you for your review. Here is a point-to-point response to each of your concerns.
>
> > Several statements in the paper are not well-justified by references, e.g. on KDE in the introduction.
>
> We believe the term curse of dimensionality and how it is associated with KDE is a term that is well known to the ICLR community. But, we are happy to add a reference to this for completeness.
>
> > More details should be added on how GAM approximates the Kolmogorov-Arnold representation and how well it performs. For example, 'based on the K-A representation theorem, GAM are able to learn the intensity of the higher-order interaction between Poisson processes' It would be nice to add references and examples since it is the key of the paper.
>
> We have provided references for the Kolmogorov-Arnold Representation Theorem; it is located on page 3 at the very bottom, just after Theorem 1 [Braun & Griebel, 2009] and [Kolmogorov, 1957]. These are the key references for this theorem. The examples can be found in the references.
>
> > Motivations for introducing APP are not well explained. Background on Poisson Processes are not clear enough and do not give any insight about PP.
>
> The theoretical motivation is shown in the second paragraph of the introduction. The practical example is shown in the third paragraph of the introduction. Our equations for the Poisson Process are well defined and provide enough context so that the entire paper is well contained and does not need additional references to understand. Of course more insights of Poisson processes can be found in our references.
>
> > The presentation of APP, in Section 2.2, should be clarified. It is not clear how to match the log-linear model (8) and the intensity defined in Sections 2 and 2.1, e.g. the taxi toy example at the top of page 5 makes things even more confusing.
>
> Do you mean top of page 4? Figure 1 shows the temporal component is along one axis, and along the other axis is the interaction effects between spatial and temporal coordinates. Equation (8) needs to be used together with Equation (7) to match the figures on the top of page 4. So the first parameter represents the interaction effect and the second parameter represents the temporal component.
>
> > I would like to see experiments on the computation time of the approach compared to existing methods as it could be an advantage of this approach.
>
> Yes, the computational time is the main advantage of this approach, especially in higher dimensions. However, it is difficult to show experimentally as other approaches are unable to be applied to any model above 4 dimensions.
>
> > The notation \mathbf{t}_i in Section 2 is very confusing.
>
>
> $t_i$ is well defined. $\mathbf{t}$ denotes the vector of events at each timestep $i$. For example, using our taxi example shown in the third paragraph of the introduction, $\mathbf{t}_{i}$ could mean $(x_i, y_i, W_i)$.
>
> > Figure 2 is cited in the text before Figure 1.
>
>
> Figures are usually numbered according to which Figure appears first on the page, not first in text.
>
> > Section 2.2 The symbol $\bot$ should be defined before, or just after, using it. Furthermore, $\bot$ denotes the least element in the partial order structure but is used before 'partial order' definition. Also, I think $J$ belongs to 2^{|[D]|}.
>
>
> $\bot$ is defined in the second paragraph of Section 2.2. Yes, $\bot$ does appear in the first paragraph, but we need to define what a partial order is before the concept of least can be explained. This occurs immediately after the partial order is defined.
>
> No, $J$ belongs to $2^{[D]}$ which denotes the powerset of $[D]$. Your suggestion $2^{|[D]|}$ denotes 2 to the power of the cardinality, which is just a natural number and $J$ cannot belong to $2^{|[D]|}$.
>
> > Equation (5) is weird since you basically inverse log-function on equation (4) but the index disappears and appears again in (6).
>
>
> Equation (5) is mathematically correct, it is just rearranging Equation (4). Equation (6) is obtained by substituting Equation (4) into Equation (1). Therefore it is a different equation that represents the log-likelihood for the $k$-th order model.
>
> > Figures are too small, it is difficult to see anything.
>
>
> There are enlarged versions available in the appendix.
>
> > "to arrive at": "to converge to"
>
>
> Thank you, we can revise this point.

---

> > ### Author Response · Authors · 2021-11-18
> > **Authors Response to Reviewer wbmT (2/2)**
> >
> > > The paper is not motivated and clear enough. Furthermore, the proposed approach is empirically poorly studied while no theoretical guarantees are investigated. Several claims are not supported by references or proof.
> >
> > We have provided the motivation of our paper in the second and third paragraph of the introduction. We have shown theoretical guarantees for our model by showing the relationship between our proposal model and the Kolmogorov-Arnold Representation Theorem as shown in Theorem 1. All our claims have all been supported by references and proofs. This is supported by the reviewers made by Reviewer s5rK. If there are any additional references or proof you would like in the paper, please mention them in the discussion phase to substantiate any claims, we are happy to include them.

---

### Official Review · Reviewer_s5rK · 2021-10-30

**Correctness:** 4
**Technical Novelty And Significance:** 3
**Empirical Novelty And Significance:** 3
**Recommendation:** 5
**Confidence:** 3

**Main Review:**

I found the main idea of the paper novel and interesting. I believe the paper is well written and easy to read even for someone not familiar with information geometry. I particularly appreciated the theoretical guarantees that come with this model and inference scheme. The experimental section is extensive but could be improved (see comments below). Addressing the following comments could help clarify the benefits and contributions of the proposed approach:

1) It seems to me that the point of this paper is not to model multiple correlated poisson processes but rather to learn the intensity of a single poisson process by “decomposing” it into sub processes over lower dimensional input spaces. Can you confirm this is correct?
2) The authors speak about general high dimensional input spaces in the introduction also making the example of space-time locations. However, the input definition is not clear in Section 2. From Section 2, the authors speak only about timestamps where each time step seems to have a D dimensionality. This is confusing. It would be useful to repeat your initial example when you define $\mathbf{t}_1, …. \mathbf{t}_N$ to give examples of what these time stamps are. Ultimately it seems to me that you could just call $\mathbf{x}_i = [x_i^1, …., x_i^{D-1}, t_i ] $ the input vector in a D dimensional space which also includes the time stamp.
3)Section 2.3. In terms of computational complexity, what is generally the cardinality of $\mathcal{S}$? Is this not given by the number of terms considered in Eq (6)? In other words the number of lower dimensional projections in Eq 4? If this is the case it does not seem the computational complexity is significantly lower compared to the methods mentioned in Section A.1 in the appendix.
4) I believe the authors should comment an important approximation introduced when discretising the input space thus getting rid of the intractable integral in the likelihood? There are indeed approaches in the literature that avoid that and despite resorting to variational inference (thus no guarantees) are scalable and seem to be working fine in experimental settings (see [1] and [2] below).
5) Figure 4. It seems like there is a typo in the captions. For the sparse case N should be 1000? Also it does not seem like the observations are sparse in the two left plots of the second row.
6) Experimental comparison. Given the number of approaches existing in the literature and using GP to model the intensity function (and same link function used in Eq (2)) I think the authors should compare the proposed method to one of them (one example could be the LGCP model) especially cause for some of them the approximated inference scheme is fast and stochastic gradient descent method can be used to reduce the computational complexity. In addition there exist out of the box tools that can be used to train such models making it a fast default choice. They would probably fail in sparse settings but I think the paper would significantly benefit from that comparison.


**Minor comments**
1) Just above Eq (6) I believe the authors refer to Eq (3) and not Eq (1)
2) The figures are very scatters in the paper and I would prefer to have them reorganised in order to have them closer to the point where they are discussed in the text. In addition it would be useful to see Fig 4 together with fig 3 and fig 5 together with fig 6 as they refer to the same experiment.

[1] Aglietti, Virginia, et al. "Structured variational inference in continuous cox process models." arXiv preprint arXiv:1906.03161 (2019).

[2] Lloyd, Chris, et al. "Latent point process allocation." Artificial Intelligence and Statistics. PMLR, 2016.


**Summary Of The Paper:**

This paper proposes a new model, called additive Poisson process, for estimating the intensity of a high dimensional Poisson process. By representing the intensity as a sum of lower dimensional projections, the method can deal with sparse data while allowing for an efficient inference scheme based on the information geometric structure of the distribution space. The assumed intensity form is motivated by the Kolmogorov-Arnold theorem while the estimation procedure has some formal guarantees of convergence.

**Summary Of The Review:**

I found the main idea of the paper novel and interesting. I believe the paper is well written and I particularly appreciated the theoretical guarantees that come with this model and inference scheme. However there are some parts of the paper that are confusing, even in the basic formulation section. Therefore, I believe the paper is borderline and could be significantly improved by clarifying the points above.

---

> ### Author Response · Authors · 2021-11-18
> **Authors Response to Reviewer s5rK (1/2)**
>
> Thank you for your positive comments! We have provided point-to-point responses to your reviews and have revised the paper accordingly based on your constructive feedback. It is appreciated if you are able to update your score based on our revised paper.
>
> > It seems to me that the point of this paper is not to model multiple correlated poisson processes but rather to learn the intensity of a single poisson process by “decomposing” it into sub processes over lower dimensional input spaces. Can you confirm this is correct?
>
> Yes, we have motivated our problem so that we have a single poisson process that can be decomposed into sub processes over lower dimensional spaces. But in theory, each of the lower dimensional spaces is a poisson process as well. So if we wish to look at the process of a particular interaction this is possible as well.
>
> > The authors speak about general high dimensional input spaces in the introduction also making the example of space-time locations. However, the input definition is not clear in Section 2. From Section 2, the authors speak only about timestamps where each time step seems to have a D dimensionality. This is confusing. It would be useful to repeat your initial example when you define $t_1, \ldots , t_N$ to give examples of what these time stamps are. Ultimately it seems to me that you could just call $x_i = [ x_i^{1}, \ldots, x_{i}^{D-1}, t_i ]$ the input vector in a D dimensional space which also includes the time stamp.
>
> This is not correct. We cannot treat $x_i = [ x_i^1, \ldots, x_i^{D-1}, t_i ]$ as an input vector in a $D$-dimensional space as the timestamp $\mathbf{t}$ and the intensity $f(\mathbf{t})$ are mixed up in the suggestion. The $D$-dimensional vector $\mathbf{t}$ in Section 2 refers to the timestamp, which corresponds to $(t_{{x_i}}, t_{y_{i}}, t_{W_{i}})$ in Introduction, and $f(\mathbf{t})$ corresponds to $(x,y,W)$.
>
>
> > Section 2.3. In terms of computational complexity, what is generally the cardinality of $\mathcal{S}$? Is this not given by the number of terms considered in Eq (6)? In other words the number of lower dimensional projections in Eq 4? If this is the case it does not seem the computational complexity is significantly lower compared to the methods mentioned in Section A.1 in the appendix.
>
> You are right that we can determine $\mathcal{S}$ according to Equation (6). In such a case, the cardinality of $\mathcal{S}$ is $O(D^k)$. Please note that how to determine $\mathcal{S}$ belongs to the model selection problem and our method offers its flexible choice. For example, if we have some domain expertise in the application, we can more appropriately select the parameters that are most important. Such flexible turing of the parameter domain cannot be achieved by the existing approaches described in Section A.1.
>
> > I believe the authors should comment an important approximation introduced when discretising the input space thus getting rid of the intractable integral in the likelihood? There are indeed approaches in the literature that avoid that and despite resorting to variational inference (thus no guarantees) are scalable and seem to be working fine in experimental settings (see [1] and [2] below).
>
> Thank you for the suggestion. We do agree that this is an important point and we have revised the paper to include this discussion in the first paragraph of Section 2.2 in our revised version.
>
> > Figure 4. It seems like there is a typo in the captions. For the sparse case N should be 1000? Also it does not seem like the observations are sparse in the two left plots of the second row.
>
> Thank you for picking this up. We have the numbers for sparse and dense the wrong way around. The sparse experiment should have 1000 points and the dense experiment should have 100,000 points. There are 1000 points in a 2 dimensional space, there are only 7 points in process [1,2].
>
> > Experimental comparison. Given the number of approaches existing in the literature and using GP to model the intensity function (and same link function used in Eq (2)) I think the authors should compare the proposed method to one of them (one example could be the LGCP model) especially cause for some of them the approximated inference scheme is fast and stochastic gradient descent method can be used to reduce the computational complexity. In addition there exist out of the box tools that can be used to train such models making it a fast default choice. They would probably fail in sparse settings but I think the paper would significantly benefit from that comparison.
>
> Thank you for your valuable suggestion. Although our objective is to examine the performance of our method compared to well-established baselines, we will consider this comparison in our future work.

---

> > ### Author Response · Authors · 2021-11-18
> > **Authors Response to Reviewer s5rK (2/2)**
> >
> > > Just above Eq (6) I believe the authors refer to Eq (3) and not Eq (1)
> >
> > Equation (3) is just a rearrangement of Equation (1). But yes, I agree with you, it is probably easier to arrive at Equation (6) if we substitute Equation (4) into Equation (3) instead of Equation (1). We have revised our paper accordingly.
> >
> > > The figures are very scatters in the paper and I would prefer to have them reorganised in order to have them closer to the point where they are discussed in the text. In addition it would be useful to see Fig 4 together with fig 3 and fig 5 together with fig 6 as they refer to the same experiment.
> >
> > Thank you for pointing this out. We have uploaded a revised version.
> >
> >
> > > I found the main idea of the paper novel and interesting. I believe the paper is well written and I particularly appreciated the theoretical guarantees that come with this model and inference scheme. However there are some parts of the paper that are confusing, even in the basic formulation section. Therefore, I believe the paper is borderline and could be significantly improved by clarifying the points above.
> >
> > We have provided clarifications for the points above. It is appreciated if you can revise the score for the review.

---

### Official Review · Reviewer_AyVi · 2021-11-02

**Correctness:** 2
**Technical Novelty And Significance:** 3
**Empirical Novelty And Significance:** 3
**Recommendation:** 5
**Confidence:** 3

**Main Review:**

•	The idea of restricting your model to lower-order interactions is well-motivated and flexible.

•	The paper is well written (I only found one typo!).

•	In synthetic data experiments, why not compare \lambda with the estimators directly?

•	In Figure 2 (a), why does it read \eta = \hat{\eta}? I thought the purpose of the figure was to show that the parameters for order-3 or -4 interactions were set to zero?

•	Could you please give a real-life example of a four-dimensional (or higher) Poisson process? It seems to me like most real-life examples are at most three-dimensional.

•	You said that KDE is too slow for d >= 4. How does it do for d = 3?

•	How was the bandwidth chosen in the taxi experiment?

•	I like how you related the GAM to your estimation problem. It is helpful to include the Kolmogorov-Arnold representation theorem to explain how the restricted functional form is qualitatively approximating the form given in Theorem 1. However, given that this relationship is not quantified, statements like “Based on the Kolmogorov-Arnold representation theorem, generalized additive models …” (pg 4) should be scaled back. There is a similar statement in the conclusion.

•	From Figure 3, one could argue that KDE is a better method than APP for two-dimensional processes, despite its sensitivity to the kernel bandwidth.

•	In Figure 4, I can’t tell which method has the best performance. It would be helpful to give a numeric comparison, perhaps some norm of \lambda - \hat{\lambda}.

•	Please define t^{j}.

•	Please justify the use of the exponential function in the functional prior on \lambda(t) in Equation (2).

•	Please give some indications of computation time.

•	Typo on pg. 9: “The third-order method is able to period better”


**Summary Of The Paper:**

This paper considers the problem of estimating the intensity function of a multi-dimensional Poisson process. Inspired by the generalized additive model, the authors propose a certain non-parametric form for the log-likelihood of the intensity function, involving multiple functions of groups of dimensions. Further, the higher-order interactions among the dimensions of the Poisson process are restricted to be within a small order k, so only functions of up to k of the dimensions are considered. Considering lower-order interactions naturally leads to a partial order, and hence the authors propose a log-linear model. For each element in the discretized state space, there is a parameter to be estimated. In turn, searching for the parameters amounts to solving a KL divergence minimization problem, which can be solved using convex optimization. Experiments on synthetic data as well as real data (New York taxis) are included. The experiments investigate the performance of the new approach, compared with three existing methods from the literature.

**Summary Of The Review:**

The idea to decompose the function f into lower-order contributions appears to be novel, and the link to GAM’s is interesting. From the empirical evaluation, I am not convinced that the APP approach is better than existing methods for d = 2, 3, while higher-dimensional applications are not motivated.

---

> ### Author Response · Authors · 2021-11-18
> **Authors Response to Reviewer AyVi (1/2)**
>
> Thank you for your positive comments! Here are the response to your comments.
>
> > In synthetic data experiments, why not compare \lambda with the estimators directly?
>
> We did this comparison. The experimental results for the comparison of \lambda with the estimator using the KL divergence is in Figure 3 and Figure 5.
>
> > In Figure 2 (a), why does it read \eta = \hat{\eta}? I thought the purpose of the figure was to show that the parameters for order-3 or -4 interactions were set to zero?
>
> The interactions are set to 0 by letting \theta = 0. This does not mean that \eta is equal to zero, p in each of the nodes are not equal to 0 (and they cannot be 0 as this only occurs if \theta = \infty, which is undefined) (see Equation (9) and Equation (10)). Therefore, applying the equation in paragraph 2 of Section 2.3 to compute \eta, we can see that the values for \eta and \hat{\eta} are non-zeros.
>
> > Could you please give a real-life example of a four-dimensional (or higher) Poisson process? It seems to me like most real-life examples are at most three-dimensional.
>
> We have given a four-dimensional poisson process as an example in the introduction. Please see the third paragraph of the introduction. Other examples include infrastructure failure prediction. Components in an infrastructure network are very complicated, and each one of these components can be modeled as a Poisson process, and the overall event of a failure could be seen as a high-dimensional process. Another example is predicting rare diseases based on a number of factors such as their symptoms. This application is high dimensional because there are many different symptoms which lead to a disease and certain combinations of symptoms are able to characterize a particular disease. We hope explaining these applications is enough motivation for you to revise your score.
>
> > You said that KDE is too slow for d >= 4. How does it do for d = 3?
>
> Since the time complexity of KDE is O(N^D), it can also be slow if N is large. We were unable to run d=3 for all our experiments, therefore we have chosen to omit them from our results section.
>
> > How was the bandwidth chosen in the taxi experiment?
>
> We have applied the rule-of-thumb bandwidth estimator approach proposed by [Silverman 1986]. The rule-of-thumb approach takes into consideration the standard deviations and the sample size when selecting the bandwidth.
>
> > I like how you related the GAM to your estimation problem. It is helpful to include the Kolmogorov-Arnold representation theorem to explain how the restricted functional form is qualitatively approximating the form given in Theorem 1. However, given that this relationship is not quantified, statements like “Based on the Kolmogorov-Arnold representation theorem, generalized additive models …” (pg 4) should be scaled back. There is a similar statement in the conclusion.
>
> Thank you for your suggestion. Will amend our writing for this part.
>
> > From Figure 3, one could argue that KDE is a better method than APP for two-dimensional processes, despite its sensitivity to the kernel bandwidth.
>
> Yes, that is correct. However, KDE is much more computationally expensive and does not scale to high dimensions. Our approach only approximates KDE.
>
> > In Figure 4, I can’t tell which method has the best performance. It would be helpful to give a numeric comparison, perhaps some norm of \lambda - \hat{\lambda}.
>
> This can be seen in Figure 5, where we show the KL divergence from \lambda to \hat{\lambda}.
>
> > Please define t^{j}.
>
> $t$ superscript has been defined between Equation (1) and Equation (2).
>
> > Please justify the use of the exponential function in the functional prior on \lambda(t) in Equation (2).
>
> $g$ can be any positive function to constrain the intensity function to be always positive. The exponential function is just a convenient choice which allows the model to be formulated as a GAM as shown in Section 2.1.
>
> > Please give some indications of computation time.
>
> The time complexity for each approach is detailed in Section 3.2. O(N^D) for KDE, O(N^2K) for DP-Beta, O(N^2) for RKHS, and O(|\Omega||S|) for our proposed approach APP.
>
> > Typo on pg. 9: “The third-order method is able to period better”
>
> Thank you, we will revise this.

---

> > ### Author Response · Authors · 2021-11-18
> > **Authors Response to Reviewer AyVi (2/2)**
> >
> > > The idea to decompose the function f into lower-order contributions appears to be novel, and the link to GAM’s is interesting. From the empirical evaluation, I am not convinced that the APP approach is better than existing methods for d = 2, 3, while higher-dimensional applications are not motivated.
> >
> > For dimensions 2 and 3, the run-time should be significantly faster as the computational complexity is O(N^D) for KDE, O(N^2 K) for DP-Beta, O(N^2) for RKHS, and O(|\Omega||S|) for our proposed approach APP. As you can see, KDE scales exponentially with respect to the number of dimensions. From the empirical evaluations, there is very little difference between the results between KDE and APP, but the computational cost is significantly lower with respect to the number of dimensions. Our future work includes applying this model to predict rare diseases based on a number of factors such as their symptoms. This application is high dimensional because there are many different symptoms which lead to a disease and certain combinations of symptoms are able to characterize a particular disease. Our approach is particularly advantageous as the lower dimensional spaces are interpretable. We hope this has given you some clarity on some potential high-dimensional applications. We hope you can revise your score accordingly.

---

> > > ### Comment · Reviewer_AyVi · 2021-11-22
> > > **Further comments**
> > >
> > > Thank you for the clarification; I had thought that the taxi example was three dimensional (x-axis, y-axis, day of the week), but I had overlooked the fact that the intensity function is inhomogeneous. Now my understanding is that the dimension in the taxi example is 4, but D = 3. It would be helpful to clarify more explicitly that D and the dimension are different.
> > >
> > > The method is stated for general D, but only evaluated for D <= 4. This is fine, but it would be helpful to tell the reader early that the intent is to use it on small D and therefore focus the evaluation on this regime. In some sense you already do this with the running example, but being more explicit would help. Otherwise, if the writing is for general D, then the evaluation should reflect this.
> > >
> > > To clarify: my suggestion of “direct comparison” of \lambda with the estimators meant comparing them as functions, using some function norm.
> > >
> > > Figure 2 (a): To clarify, I don’t understand the explanatory purpose of writing \eta = \hat{\eta}.
> > >
> > > The proposed methodology is interesting, but the empirical evaluation is a weakness in my opinion. First, empirical execution times are missing. Second, it seems that an important comparison was not included; reviewer cFZz raised the point that the approach should be compared with the GAM + tensor product bases.

---

> > > > ### Author Response · Authors · 2021-11-23
> > > > **Response to Further Comments by Reviewer AyVi**
> > > >
> > > > Thank you for taking the time to respond to our comments. This is much appreciated. As you can see, there are very few approaches which can be applied to the problem that we are solving. Below we will provide further clarification to your comments.
> > > >
> > > > > Thank you for the clarification; I had thought that the taxi example was three dimensional (x-axis, y-axis, day of the week), but I had overlooked the fact that the intensity function is inhomogeneous. Now my understanding is that the dimension in the taxi example is 4, but D = 3. It would be helpful to clarify more explicitly that D and the dimension are different.
> > > >
> > > > Yes, that is correct. We will revise the wording of this point in the camera ready version to make sure the reader understands that the intensity function is inhomogenous.
> > > >
> > > > > The method is stated for general D, but only evaluated for D <= 4. This is fine, but it would be helpful to tell the reader early that the intent is to use it on small D and therefore focus the evaluation on this regime. In some sense you already do this with the running example, but being more explicit would help. Otherwise, if the writing is for general D, then the evaluation should reflect this.
> > > >
> > > > Yes, in our experiments we have only evaluated for D <= 4, as we are able to more strongly present our results. Our approach can certainly be applied for a general D, however, we do not believe that presenting our results in higher dimensions provides any additional insights as it is difficult to understand the results from higher dimensions.
> > > >
> > > > > To clarify: my suggestion of “direct comparison” of \lambda with the estimators meant comparing them as functions, using some function norm.
> > > >
> > > > Using some function norm to compare \lambda with the estimator is certainly possible. Although we think that evaluation on the KL divergence is more meaningful, for the camera-ready version, we are happy to provide the norm between the two functions in the appendix.
> > > >
> > > >
> > > > > Figure 2 (a): To clarify, I don’t understand the explanatory purpose of writing \eta = \hat{\eta}.
> > > >
> > > > Figure 2(a) is a visualization of the entire outcome space (the power set). We select a subset of parameters to be included into the model, to which we are using first order moment matching in optimization. \eta = \hat{\eta} is to show the parameters that we are using for first order moment matching.
> > > >
> > > > > The proposed methodology is interesting, but the empirical evaluation is a weakness in my opinion. First, empirical execution times are missing. Second, it seems that an important comparison was not included; reviewer cFZz raised the point that the approach should be compared with the GAM + tensor product bases.
> > > >
> > > > In our current submission we have already provided the theoretical analysis on the run-time executions of each of the approaches. Our proposed approach contains a lot of free parameters such as M and S which have a huge impact on the run-time, it may be difficult to provide a meaningful insight using the empirical execution time.
> > > >
> > > > As mentioned in our response to reviewer cFZz, our approach is not exactly the same as the GAM + tensor product based approaches. Though it may be possible to provide this comparison, on the surface level, it is not exactly clear on how to formulate the GAM + tensor product based approaches to our current experiments.

---

### Official Review · Reviewer_cFZz · 2021-11-04

**Correctness:** 3
**Technical Novelty And Significance:** 2
**Empirical Novelty And Significance:** 2
**Recommendation:** 3
**Confidence:** 4

**Main Review:**

The paper is overall well written and I read it with interest. The major issue I have with this paper is that the proposed approach has long existed in the statistics literature for decades, under the context of generalized additive models. In particular, in chapter 5.6 of  [1], the tensor product bases are more flexible than the approach proposed in this paper. In my opinion, the lower-dimensional approximation proposed in this paper is essentially a piecewise constant approximation of the intensity function (in lower dimension), which is a special case of the tensor product basses.  If comparisons should be made, it should be between the proposed method and the GAM with tensor product bases, instead of the RKHS formulation.

In addition, the partial order in the formulation (7) seems to rely on the assumption that all dimensions have the same common support [0, T], which is not the case for the spatial-temporal processes considered in this paper. How is this issue addressed?

For the proposed method, the most difficult task should be to determine the value of hyper-parameters M and h. If M is too large, it can easily lead to overfitting. If M is too small, there will be a large bias in the estimated intensity functions. Please provide more details on how cross-validation is performed to choose an optimal combination of h and M.

Another difficult task with the proposed method is how to justify the use of lower-dimensional approximation. For example, is it justifiable to assume that there is no need to consider the spatial and temporal interaction in a spatial-temporal point process? In GAM, one can do this by performing an analysis of variance (ANOVA), see, e.g., section 5.6.3 of [1]. Could this be done with the proposed approach?

Minor points:
1. In the motivating example, the time coordinate is the day of the week, which is discrete. Such an example is not coherent with the definition of the Poisson process, which should be with continuous coordinates.
2.  in equation (6), should it be $t_i^{(I)}$ instead of $t^{(I)}$ inside the function f_{I}?

Reference:

[1]. Wood, S.N., 2017. Generalized additive models: an introduction with R. CRC Press. (2nd Edition)

**Summary Of The Paper:**

This paper proposes a method to estimate the multi-dimensional intensity function of a Poisson process with a lower-dimensional projection. The proposed method is motivated by perspectives of information geometry and generalized additive models in statistics. The performance of the proposed method is extreme through synthetic examples and a real data analysis.

**Summary Of The Review:**

I think the paper is well written, but I do not see the methodology proposed to be better than the existing ones.

---

> ### Author Response · Authors · 2021-11-18
> **Authors Response to Reviewer cFZz (1/2)**
>
> Thank you for your constructive feedback! We will provide a point to point response to each of the comments so that you are able to clearly see the advantages of our proposed approach.
>
> > The major issue I have with this paper is that the proposed approach has long existed in the statistics literature for decades, under the context of generalized additive models. In particular, in chapter 5.6 of [1], the tensor product bases are more flexible than the approach proposed in this paper
>
> We disagree with this comment. Our proposed approach has advantages over some of the existing approaches as it provides a convenient graph structure in the form of a partial order to systematically allow the interactions between the features and time to be represented. Such modeling is not possible in existing approaches. The decomposition into lower dimensional space is meaningful as it represents a poisson process for each of the features in the interaction effects between features.
>
> > In my opinion, the lower-dimensional approximation proposed in this paper is essentially a piecewise constant approximation of the intensity function (in lower dimension), which is a special case of the tensor product basses. If comparisons should be made, it should be between the proposed method and the GAM with tensor product bases, instead of the RKHS formulation.
>
> Thank you for your interesting insight. Our intensity function is essentially given by Equation (8), and this is not piecewise constant. So we are struggling to find the connection between piecewise constant approximation and our formulation. We are happy to answer your question if you could elaborate a bit more about your suggestion.
>
> Our comparison with RKHS is still meaningful as it is a well established baseline known by many people in the ICLR community.
> > In addition, the partial order in the formulation (7) seems to rely on the assumption that all dimensions have the same common support [0, T], which is not the case for the spatial-temporal processes considered in this paper. How is this issue addressed?
>
> All dimensions having the same support is not a problem. T represents the largest value in any of the dimensions, while 0 being the smallest in any dimension. If the range of the support is a concern, it can be normalized between [0, T]. But theoretically there should be no difference in performance whether the support is normalized or not.
>
> > For the proposed method, the most difficult task should be to determine the value of hyper-parameters M and h. If M is too large, it can easily lead to overfitting. If M is too small, there will be a large bias in the estimated intensity functions. Please provide more details on how cross-validation is performed to choose an optimal combination of h and M.
>
> The selection of the number of bins M is highly application dependent. It determines the time interval where the events are considered to have interacted with each other. In our synthetic experiment, we have defined that there is an interaction when they are within 0.1sec apart. So a reasonable choice for M in our model is M = T/dt where dt = 0.1sec and T is max time. A larger M means that there is a smaller time interval where the events in different stochastic processes are able to interact. Likewise, rule of thumb or cross validation is recommended to select M in practical applications.
>
> The bandwidth h in Table 1 is selected through grid search. We tried a range of bandwidth and have selected the bandwidth with the lowest KL divergence between KDE and the latent intensity function. We understand this is only possible in synthetic data and not possible to be done in the real-world situation since the latent intensity function is unknown. Our experiment is designed to demonstrate the sensitivity of the bandwidth. For practical applications, we suggest using conventional bandwidth approaches such as rule of thumb or cross validation depending on the application of the model.
>
>
> > Another difficult task with the proposed method is how to justify the use of lower-dimensional approximation. For example, is it justifiable to assume that there is no need to consider the spatial and temporal interaction in a spatial-temporal point process? In GAM, one can do this by performing an analysis of variance (ANOVA), see, e.g., section 5.6.3 of [1]. Could this be done with the proposed approach?
>
> We have not made the assumption that there are no interactions between the spatial and temporal interactions. In fact spatial-temporal interactions can be included into the model depending on the design of the graph structure. This can certainly be done with our proposed approach.

---

> > ### Author Response · Authors · 2021-11-18
> > **Authors Response to Reviewer cFZz (2/2)**
> >
> > > In the motivating example, the time coordinate is the day of the week, which is discrete. Such an example is not coherent with the definition of the Poisson process, which should be with continuous coordinates.
> >
> > You are correct that the day of the week is a discrete variable. But, a continuous time component t is associated with each (x, y, W) as we have introduced in Introduction. So in fact in this example there are two spatial coordinates and two temporal coordinates, a discrete one which represents the day of the week and a continuous one which represents the time in each day.
> >
> > > in equation (6), should it be $t_{i}^{(I)}$ instead of $t_{i}$ inside the function $f_{I}$?
> >
> > No, Equation (6) is correctly defined. $\mathbf{t}$ with a superscript is defined between Equation (1) and Equation (2). The superscript represents the set of processes. $\mathbf{t}$ with a subscript is defined just above Equation (1) which defines the events at each timestep. We have not defined a notation for $\mathbf{t}$ that contains both the superscript and subscript.
> >
> > > I think the paper is well written, but I do not see the methodology proposed to be better than the existing ones.
> >
> > In our response we have shown that our approach does have advantages and it is formulated differently to the existing approaches. We hope you can see the advantages of our approach and revise your score accordingly.

---

> > > ### Comment · Reviewer_cFZz · 2021-11-24
> > > **Response to response**
> > >
> > > 1. "We disagree with this comment. Our proposed approach has advantages over some of the existing approaches as it provides a convenient graph structure in the form of a partial order to systematically allow the interactions between the features and time to be represented. Such modeling is not possible in existing approaches. The decomposition into lower dimensional space is meaningful as it represents a poisson process for each of the features in the interaction effects between features."
> > >
> > > Response: I am not saying decomposition into lower dimensional space is not meaningful. I was saying that this can be done using existing tools such as tensor product bases.
> > >
> > > 2. "Our comparison with RKHS is still meaningful as it is a well-established baseline known by many people in the ICLR community."
> > >
> > > Response: anything you can do with RKHS, you can also do it with tensor product bases, which is very close to RKHS.
> > >
> > > 3. The problem of selecting the M and h is not addressed. The synthetic experiments should adopt the same strategy as those will be used in real applications for an honest assessment of the proposed approach.
> > >
> > > 4. "You are correct that the day of the week is a discrete variable. But, a continuous-time component t is associated with each (x, y, W) as we have introduced in the Introduction. So in fact in this example, there are two spatial coordinates and two temporal coordinates, a discrete one which represents the day of the week and a continuous one which represents the time in each day."
> > >
> > > Response: Regardless of it is four-dimension or three-dimension, having a discrete coordinate dimension violates the basic assumption of a Poisson process.
> > >
> > > 5. If there is no problem with equation (6), where is the summation index $i$  reflected in the summands?

---

> > > > ### Author Response · Authors · 2021-11-25
> > > > **Response to Further Comments by Reviewer cFZz**
> > > >
> > > > Thank you for your valuable comments. Here is the response to your concerns.
> > > >
> > > > > The problem of selecting the M and h is not addressed. The synthetic experiments should adopt the same strategy as those will be used in real applications for an honest assessment of the proposed approach.
> > > >
> > > >
> > > > Our technique to select M and h for our synthetic experiments can be applied to real-world applications. In our synthetic experiment, we are able to select M because we know the grid size of the underlying generative process. Therefore in our synthic experiment we have chosen M to be M=T/dt, where dt=0.1 and T is the maximum time. Likewise, in real-world applications some domain expertise can be used to identify what is a good window length to capture the events we are interested in. If we do not have any domain expertise on a particular dataset, we are able to use an empirical approximation by looking at the dataset and identify the smallest interval between events and select the inverse of the smallest interval multiplied by the maximum time to be M (rounded up). This approach to selecting M in the real-world applications is the same as synthetic experiments.
> > > >
> > > > We understand that h is much harder to select using real-world applications, that is why we have Figure 3 and Figure 5 demonstrating the effects of varying the bandwidth. We understand Figure 3 and Figure 5 is only possible if we have the true intensity function which is impossible to obtain in real-world applications. But our objective in the synthetic experiment is to demonstrate the sensitivity of the bandwidth. So therefore in our real-world application, we have chosen to use a heuristic based approach such as Silverman’s rule of thumb approach or cross validation (if possible) is used to select the bandwidth. I hope this has addressed your concerns regarding selecting M and h.
> > > >
> > > > > Regardless of it is four-dimension or three-dimension, having a discrete coordinate dimension violates the basic assumption of a Poisson process.
> > > >
> > > >
> > > > This is not correct. Only t needs to be continuous for it to be a Poisson process. (x, y, w) is taken into account by f(t), where f can be a function which takes input of discrete or continuous variables. The basic assumption of a Poisson process has not been violated.
> > > >
> > > > > If there is no problem with equation (6), where is the summation index i reflected in the summands?
> > > >
> > > > I see, you are correct. Thank you for pointing this out. We will update the notation to have $\mathbf{t}_i^{I}$ defined.

---

### Decision · Program_Chairs · 2022-01-20

**Decision:**

Reject

**Comment:**

The paper proposes a novel approach for estimating the high-dimensional intensity function of a Poisson process. The proposed approach builds on generalized additive models, using lower-dimensional projections.

The reviewers noted that, although the paper is well written, the position of this paper compared to earlier related work is unclear, and the empirical evaluation of the method should be strenghtened. The authors clarified some points in their response, but the paper would still require some more modifications to be ready for publication. I therefore recommend this paper to be rejected.